# Calprotectin in Lung Diseases

**DOI:** 10.3390/ijms22041706

**Published:** 2021-02-08

**Authors:** Ourania S. Kotsiou, Dimitrios Papagiannis, Rodanthi Papadopoulou, Konstantinos I. Gourgoulianis

**Affiliations:** 1Department of Respiratory Medicine, Faculty of Medicine, University of Thessaly, 41110 Larissa, Greece; kgourg@med.uth.gr; 2Department of Nursing, Faculty of Medicine, University of Thessaly, 41110 Larissa, Greece; dpapajon@uth.gr; 3Human Nutrition, School of Medicine, College of Medical Veterinary and Life Sciences, University of Glasgow, Glasgow Royal Infirmary, Glasgow G31 2ER, UK; rodanthi.papadopoulou@gmail.com

**Keywords:** biomarker, calgranulin, calprotectin, lung, S100A8/A9

## Abstract

Calprotectin (CLP) is a heterodimer formed by two S-100 calcium-binding cytosolic proteins, S100A8 and S100A9. It is a multifunctional protein expressed mainly by neutrophils and released extracellularly by activated or damaged cells mediating a broad range of physiological and pathological responses. It has been more than 20 years since the implication of S100A8/A9 in the inflammatory process was shown; however, the evaluation of its role in the pathogenesis of respiratory diseases or its usefulness as a biomarker for the appropriate diagnosis and prognosis of lung diseases have only gained attention in recent years. This review aimed to provide current knowledge regarding the potential role of CLP in the pathophysiology of lung diseases and describe how this knowledge is, up until now, translated into daily clinical practice. CLP is involved in numerous cellular processes in lung health and disease. In addition to its anti-microbial functions, CLP also serves as a molecule with pro- and anti-tumor properties related to cell survival and growth, angiogenesis, DNA damage response, and the remodeling of the extracellular matrix. The findings of this review potentially introduce CLP in daily clinical practice within the spectrum of respiratory diseases.

## 1. Introduction

Calprotectin (CLP) is a heterodimeric complex formed by two binding proteins of the calcium ion, which belong to the S-100 protein family, S100A8 and S100A9, having both anti-inflammatory and anti-bacterial properties. The first applied name to CLP was major leukocyte protein L1 or 27E10 [1,2,3]. It was later recognized as the combination of S100A8 and S100A9, while various synonyms have also been adopted, such as myeloid-related proteins-8 and -9 (MRP-8 and MRP-9), calgranulin A and B, or migration inhibitory factor-related protein of 8 and 14 kDa (MRP-8 and MRP-14), respectively [1,2,3].

CLP represents almost two-thirds of the soluble cytosolic protein content of neutrophils. CLP may also be detected at various levels in monocytes, macrophages, epithelial cells, and platelets [4,5]. It is expressed as an anti-bacterial agent, mainly but not exclusively by neutrophils, when activated [3]. Upon cell activation or death, CLP is released extracellularly, acts as an alarmin, or damage-associated molecular pattern (DAMP). It serves as a stimulatory ligand for the innate cell-surface receptors such as receptor for advanced glycation end products (RAGE) and Toll-like receptor 4 (TLR4) [4,5]. However, there are data demonstrating that physiological levels of S100A9 homodimers can trigger an inflammatory response in vivo despite the capacity of RAGE and TLR4 blockade to inhibit responses in vitro [6]. CLP inhibits the oxidative metabolism of polymorphonuclear neutrophils in vitro, an effect that can be potentiated by the controlled activation of the protease-activated receptor-2 (PAR2) [7]. In particular, the survival rate was almost doubled from 33% to 65% or 63% in mice treated with S100A8 or PAR2 activating peptide, respectively, whereas 85% of the mice were treated with both PAR2 activating peptide and S100A8 survived, at a statistically significant higher rate than those treated with a single agent [7].

CLP has an important epithelial barrier function, and proper levels of the heterodimer may be needed for both immune defense and homeostasis [8,9]. CLP has microbicidal, cytotoxic functions via heavy-metal detoxification [10]. CLP plays a critical role in multiple cellular processes, including cell cycle progression, proliferation, differentiation, chemotaxis, migration in a calcium-dependent manner, and survival, as well as in redox regulation, proteins’ phosphorylation, and cytoskeletal components rearrangement [5,10,11,12]. The features that CLP possesses, such as its small size, easy diffusion between tissue and blood, and enzymatic degradation resistance, make it a sensitive marker of neutrophil activation anywhere in the body [11,12].

High levels of CLP have been found in many types of infectious and inflammatory diseases, including sepsis [13], inflammatory bowel disease, myocardial infarction, and rheumatological diseases, where they are closely associated with disease severity [13,14,15,16,17,18]. The implication of S100A8/A9 in the inflammatory process was shown more than 20 years ago; however, its role in the pathogenesis of respiratory diseases or its usefulness as a biomarker for the appropriate diagnosis and prognosis of lung diseases have gained increasing attention in recent years. This review aims to provide current knowledge regarding the potential role of CLP in the pathophysiology of lung diseases and describe how this knowledge is translated into daily clinical practice.

## 2. The Expression of CLP in Lung Tissue and Potential Mechanisms of Extracellular Release of Human Phagocyte CLP

It has been previously shown that S100A9 is not basally present in the healthy airway epithelial cells but is highly expressed in type II pneumocytes in alveoli [8,9,19]. However, a study found that after an *Alternaria* exposure, S100A9 expression was high both in the alveolar and bronchial epithelium as a response to local environmental stress [9]. Similarly, S100A8 and S100A9 protein secretion was stimulated in the secretions of human bronchial epithelial cells and primary human bronchial epithelial cells after lipopolysaccharide stimulation [20]. Intranasal administration of murine S100A9 adenovirus induced a time-dependent influx of macrophage that coexisted with increased S100A9 levels and pro-inflammatory cytokines in the bronchoalveolar lavage fluid (BALF) [6]. Conversely, other authors found no increase of CLP in healthy human peripheral airways after local endotoxin exposure [21].

A much-debated issue is how fast CLP is released from cells in response to encounters with bacteria or PAMPs. In animals with severe bacterial infection, CLP elevation has been described after two days [22]. Another animal study described similar findings a few hours after endotoxin challenge [21]. Activation and differentiation of alveolar monocytes into MRP-8/14-positive mature macrophages with the progress of asphyxia duration have been detected in specimens of suffocated human lungs [23]. Human data on the kinetics of CLP are scarce. Van Zoelen et al. described early changes to CLP levels in healthy volunteers after endotoxin challenge with very low endotoxin doses [24]. Lipsey et al. found that although an early phenomenon, the release of CLP might not be a first line of bacterial or endotoxin inflammatory response [25]. In earlier stages of neutrophil activation, neutrophils underwent degranulation whilst keeping the integrity of their cell membrane. In this process, they would release their granular contents, such as gelatinase, myeloperoxidase (MPO), and Oncostatin M (OSM), which have also potential roles in inflammation and infection [25].

In long alveolar asphyxia cases, a 2-fold up to a 4-fold increase of MRP-14/MRP-8 was shown compared to short asphyxia [23]. In freshly isolated monocytes during prolonged cultivation, a strong upregulation of CLP was only observed during the first 3–4 days, and then declined [26]. MRP-8 (S100A8) and MRP-14 (S100A9) were expressed at defined stages of monocyte/macrophage differentiation and specifically at the stage between the resting, circulating blood monocytes and the mature tissue macrophage [26]. At the same time, while MRP-14 was expressed to varying degrees by intravascular monocytes and perivascular macrophages in acute and chronic inflammation, MRP-8 was only detected in the macrophages of chronic but not acute inflammatory tissues [26]. These data provide evidence implicating the involvement of additional triggers accelerating the recruitment of macrophages under various inflammatory conditions [26] and support that chronic inflammation is mirrored by the presence of MRP-8-positive macrophages in the tissue. Hence, it seems that the characterization of subpopulations within the alveolar macrophages may be a useful tool for discriminating inflammatory statutes and monitoring disease progression. Alongside this, it has been demonstrated an extremely high increase of MRP-8/14- levels in traumatic deaths due to extreme stress and an immediate release of catecholamines and further mediators at maximal levels within seconds [27,28,29]. Recently, distinct exercise-intensity dependent changes in serum CLP following various types of extreme physical exertion in healthy volunteers have been reported [30].

Another major topic is how CLP is released from neutrophils or other cells where it is found, including monocytes and macrophages. Although it has been proposed that S100A8/A9 extracellular release exclusively correlates with neutrophil necrosis, or disruption of the neutrophil cell membrane, more recent data have demonstrated that CLP is secreted either via distinct mechanisms of secretion or following the activation of different signal transduction pathways from undamaged cells [31].

CLP is secreted by resting and stimulated neutrophils [32]. S100A8/A9 has been reported in granules [33], suggesting that it could be released following neutrophil degranulation [33]. Hetland et al. previously reported the release of S100A8/A9 by neutrophils stimulated with N-Formylmethionyl-leucyl-phenylalanine (fMLP), a potent inducer of degranulation [34]. Nevertheless, recent evidence suggests that degranulation is not involved in the secretion of S100A8/A9 from neutrophils [32]. Calcium mobilization induced by fMLP could activate the translocation of S100A8/A9 to the plasma membrane or cytoskeleton [35,36], which could explain the disappearance of S100A8/A9 in the cytosol reported by Hetland et al. although this translocation is not sufficient to allow its secretion. Other signals are therefore necessary to induce the secretion of calgranulins. Another possible route of secretion for calgranulins is vesicular secretion, which depends on intracellular membrane-bound intermediates that need to fuse with plasma membranes to release cargo into the extracellular space. Such mechanisms involve either secretory lysosomes, exosomes derived from multivesicular bodies, or microvesicle shedding from cell surfaces [37]. Urban et al. supported that the heterodimer CLP is released in neutrophil extracellular traps (NETs) as the major anti-microbial component [38]. According to the authors, NET formation is a mechanism that ensures the interaction between cytoplasmic CLP and extracellular microbes at high local concentrations [38]. Monocytes secrete S100A9 and S100A8/A9, but not S100A8 alone, upon stimulation with pokeweed mitogen [39]. Moreover, a non-classical and tubulin-dependent secretion mechanism was shown in monocytes activated by stress [40] and inflammatory cytokines [41]. Besides, S100A9 has been found to be actively released to the extracellular environment via DDX21–TRIF signaling from undamaged macrophages, resulting in an exaggerated lung inflammatory response and cell death during influenza infection. Activation of protein kinase C by pro-inflammatory stimuli and elevation of intracellular [Ca^2+^] following contact with activated endothelium, collagen, or fibronectin can also stimulate S100A8/A9 release from phagocytes [18,42,43,44].

## 3. CLP’s Concentration Variations

The normal serum levels of CPL are in the range of 1–6 mg/L, and increased more than 100-fold with active inflammation [5]. No significant gender differences in serum CLP levels have been reported [45]. On the other hand, age-related serum CLP level differences have been documented [46]. The serum CLP reference interval for healthy elderly people from *75* years old and above has been estimated from 0.3 to 2.6 mg/L [45]. Serum CLP is particularly decreased in elderly patients with chronic rhinosinusitis [47]; however, it increases in other disease statutes, such as rheumatoid arthritis or other rheumatological diseases [48,49]. Epithelial expression of S100A8/A9 is partly regulated by the IL-6 trans-signaling pathway [47], which may be inhibited by the increased soluble gp130 (sgp130) in elderly patients with chronic rhinosinusitis and nasal polyps [47]. In that context, restoration of barrier function by targeting sgp130 has been proposed as a novel treatment strategy for this group of patients [47]. Moreover, serum CLP is higher in obese and overweight children than in normal-weight subjects [50]. An association between increased CLP and obesity has also been reported for adults [50].

Most studies report the normal range of fecal CLP to be 10 to 50 mg/L [5]. In children with cystic fibrosis (CF), fecal CLP levels were age-dependent [46]. Fecal CLP levels above 50 mg/kg were acknowledged to be elevated in children between 4 to 10 years of age, while in children less than 4 years, CLP levels were difficult to be interpreted as they are naturally high in healthy infants [46]. Fecal CLP levels demonstrated an upward trajectory until 4 years [51]. Another study reported that children aged 2–9 years had significantly higher fecal CLP concentrations than subjects aged above 10 [52]. Adults aged 60 years or older had a higher fecal CLP concentration than those aged 10–59 years [52].

Hence, careful interpretation of serum or fecal CLP levels is required if used in drug trials, particularly in children less than 10 and especially less than 4 years, and older adults over the age of 60.

## 4. CLP in Respiratory Infections

In pneumonia patients, CLP levels are elevated in BALF, lung tissue, and serum [53,54]. Moreover, calgranulins exist in the heterodimeric form in secretions of pneumonia patients in vivo [55]. Interestingly, serum CLP emerges as both a potential early marker of bacterial etiology and a predictor for five-year all-cause mortality in community-acquired pneumonia [56,57]. Messenger RNA (mRNA) sequencing from the peripheral blood of patients with pneumonia revealed that S100A9 might contribute solely to mild pneumonia [56].

CLP is a major component of neutrophils that is released upon infection or injury. It is essential for protective immunity during infection by a variety of micro-organisms through its capacity to chelate a number of first-row transition metals, including manganese Mn(II), iron Fe(II), and zinc Zn(II), withholding these essential nutrients from microbes [53,54]; thus inhibiting microbial growth [5]. The ability of CLP to affect Mn(II) availability to microbes was first recognized during studies of murine tissue abscesses infected with the Gram-positive opportunistic human pathogen Staphylococcus aureus [58] and later examined for the Gram-negative gastrointestinal pathogen Salmonella enterica serovar Typhimurium [59]. More recently, the metal competition between human CLP and bacterial metal transport machinery was evaluated, and it was found that CLP outcompetes bacterial Mn(II)-acquisition proteins under conditions of high carbonic anhydrase II, as found in the extracellular environment [60]. It is important to note that Mn, Fe, and Zn acquisition systems are viable therapeutic targets to combat multidrug-resistant microbial infections [61]. In sharp contrast to the reported host-protective role of CLP in several infections, a previous study reveals that in a model of community-acquired pneumonia, CLP is misused by Streptococcus pneumoniae, facilitating bacterial growth by attenuating Zn toxicity toward the pathogen [54].

Μost studies highlighted that CLP is strongly involved in the transepithelial migration of neutrophils and macrophages to the alveoli in streptococcal pneumonia [57,62]. CLP contributes to the host response to pneumococcal infection by increasing circulating neutrophils principally by regulating granulocyte colony-stimulating factor (G-CSF) production [62]. Mainly in pneumonia, compared to chronic obstructive pulmonary disease (COPD) or idiopathic pulmonary fibrosis, alveolar macrophages are characterized by an increased MRP-8/MRP-14 expression [63].

Staphylococcus aureus pneumonia has been associated with a substantial CLP rise in BALF and lung tissue [57]. CLP serves in an unexpected protective role for the lung in staphylococcal pneumonia [57]. MRP-14 deficiency effected Staphylococcus clearance and was associated with increased cytokine levels and diminished transmigration of neutrophils into BALF at late time-points after infection, together with reduced release of nucleosomes [57].

In the case of Acinetobacter baummani pneumonia, it has been reported that CLP is detectable within six hours of infection as immune cells respond to the invading pathogen and as the bacterial burden decreases, signals from the CLP decrease [61,64]. CLP inhibits Acinetobacter baumannii growth in vitro through the chelation of Mg (II) and Zn(II). Zn limitation reverses carbapenem resistance in multidrug-resistant Acinetobacter baumannii underlining the clinical relevance of increased CLP activity [61]. Consistent with the in vitro data, imaging mass spectrometry revealed that CLP accompanies neutrophil recruitment to the lung and accumulates at foci of infection in a murine model of Acinetobacter baumannii pneumonia [61].

CLP is also a key player in protective innate immunity during Klebsiella pneumonia [54]. Moreover, elevated levels of CLP detected by ELISA have been reported in the case of Burkholderia mallei infection, another Gram-negative, bipolar, aerobic bacterium, which is complicated with skin and lung abscesses [65]. CLP has also been used as a marker of alveolar NET formation, NETosis, and neutrophilic inflammation in ventilator-associated pneumonia [66]. S100A8/A9 proteins increase during lung injury and contribute to inflammation induced by high tidal volume mechanical ventilation combined with lipopolysaccharide. Nevertheless, in the absence of lipopolysaccharide, high levels of extracellular S100A8/A9 still amplify ventilator-induced lung injury via TLR4 [67]. S100A8 was found to induce IL-10 in vivo, initiating a feedback loop that attenuates acute lung injury [68].

In children, CLP was found significantly elevated in the sputum of patients with bronchiolitis obliterans as a marker of ongoing neutrophilic inflammation [69]. Some studies provide evidence for a transient inflammation in the lung characterized by very early recruitment of neutrophils associated with high expression levels of S100A8 and S100A9 proteins in filarian infections [70]. Besides, CLP plays a critical role in regulating Leishmania infection [71]. The rapid secretion of CLP by neutrophils at the site of infection protects uninfected macrophages and favors a more efficient innate inflammatory response against Leishmania infection, revealing how CLP can subvert this pathogen’s action [71].

CLP is released during mycobacterial infection in vitro and in vivo [72]. Studies revealed that serum S100A9 levels are significantly increased in tuberculosis than in other lung diseases [73], suggesting that S100A9 may be a potential diagnostic biomarker for pulmonary tuberculosis [73,74]. It has also been proposed that targeting S100A8/A9 proteins can decrease lung tissue damage without impacting protective immunity against tuberculosis [75]. Furthermore, S100A9 has been associated with tuberculosis development and can distinguish between different disease stages [76].

A major pathologic role for S100A8/A9 proteins in mediating neutrophil accumulation and inflammation associated with tuberculosis has been published. CLP-positive neutrophils were abundant in regions adjacent to the caseum, supporting the concept that granulomas have organized microenvironments that balance anti-microbial, anti-inflammatory responses to limit pathology in the lungs [75]. S100A9 has been strongly expressed in all granulomatous conditions, while S100A8 has been variably expressed. An old study shows differences in immunophenotype between non-phagocytic mononuclear phagocytes in epithelioid giant cells granulomas without necrosis due to delayed cell-mediated immune reactions and phagocytic mononuclear phagocytes in caseating granulomas [77]. Mononuclear phagocytes in granulomas of foreign body type, cat-scratch disease, and erythema nodosum strongly expressed S100A8 [78]. In contrast, S100A8 expression was weak or absent in mononuclear phagocytes of sarcoidosis and tuberculosis [78].

Regarding multiorgan tuberculosis, it has been shown that high fecal CLP levels could differentiate intestinal from pulmonary tuberculosis [68]. High fecal CLP levels in intestinal tuberculosis were associated with granulomas in intestinal biopsies [79]. Patients with combined pulmonary and intestinal tuberculosis had the highest serum CLP (6.5 mg/L) and presented more severe disease [80]. The functional impairment of interaction between Zn finger genes and interferon-stimulated genes, along with a higher expression of S100A8/A9 genes, possibly form the genomic basis of tuberculosis-associated immune reconstitution inflammatory syndrome (IRIS) in a subset of patients with human immunodeficiency virus (HIV) while on highly active antiretroviral therapy [80,81]. 

MRP-8/14 has also been implicated in the autophagy-mediated elimination of intracellular Mycobacterium bovis by promoting reactive oxygen species generation, which may provide a promising therapeutic target for tuberculosis and other intracellular bacterial infectious diseases [78].

During the challenging global times of coronavirus disease 2019 (COVID-19), serum CLP levels have been found to track closely with current and future COVID-19 severity and in-hospital mortality [82], be positioned as an early indicator of respiratory failure [83], and therefore immunomodulatory treatment. These observations implicate neutrophils as potential perpetuators of inflammation and respiratory compromise in COVID-19 [84]. Moreover, patients with COVID-19-associated thrombosis had significantly higher blood levels of NETs and neutrophil activation markers such as CLP compared with COVID-19 patients without thrombosis [85]. Additionally, many studies support that fecal CLP indicates intestinal inflammation in COVID-19 [86,87,88].

Overall, the existing evidence supported that CLP plays an important role in eliminating microbes via leukocyte and macrophage migration at early and late time points after infection and potentially via preventing microbial acquisition of the essentials first-row transition metals. CLP has been suggested as a diagnostic and prognostic indicator of several microbial infections in daily clinical practice, and it is timidly recognized as a promising therapeutic target.

## 5. CLP in Serous Effusions

Discriminating between malignant pleural effusions (PEs) and non-malignant PEs remains a challenge. Generally, studies examining CLP levels in effusions are very scarce. Emerging data support that pleural CLP levels were significantly higher in malignant PEs than non-malignant ones (transudates, parapneumonic, or tuberculous PEs) [89,90,91]. In a cohort of 425 patients, pleural CLP levels ranged from 772.48 to 3163.8 ng/mL (median: 1939 ng/mL) in malignant PEs, and 3216–24,000 ng/mL in non-malignant PEs (median: 9209 ng/mL) [90]. For a cut-off value of ≤6233.2 ng/mL, a 96% sensitivity and 60% specificity were reported in discriminating malignant from non-malignant PEs [90], suggesting pleural CLP as a novel and useful diagnostic biomarker in patients with PE of uncertain etiology [90,92].

Davidson et al. reported CLP levels ranging from 301 to 9431 ng/mL (median 1155 ng/mL) in effusions of breast cancer patients [93]. In PE of patients with advanced-stage ovarian carcinoma, Ødegaard et al. described values from 1035 to 4814 ng/mL (median, 1867 ng/mL) [94]. It is important to note that a different ELISA assay was used to detect antibodies against CLP among all these studies. Although CLP constitutes an independent predictor of malignant PEs according to several studies [89,91], CLP effusion concentrations did not correlate with survival in different types of malignancy, such as ovarian carcinoma, borderline cancers of the ovary, or breast carcinoma [94].

There is only one study in a cohort of 137 participants that is based on proteomic analysis, reporting that pleural fluid levels of CLP in parapneumonic PEs were significantly higher than those in non-parapneumonic PEs (transudates, other exudates, and malignant effusions). Notably, this study found that pleural CLP was significantly elevated in patients with complicated parapneumonic PE compared to uncomplicated parapneumonic PEs [95].

Ødegaard et al. also described values from 2112 to 3431 ng/mL (median, 2581 ng/mL) for peritoneal effusions [94]. A recent study reported CLP levels in ascites samples from cirrhotic patients, ranging from 5 to 111,480 ng/mL (median, 52 ng/mL) and in malignant ascites (caused by pancreatobiliary, hepatocellular, or renal cell cancer) from 47 to 2596 ng/mL (median, 401 ng/mL) [94]. The ratio of ascites CLP to total protein has been proposed as a promising biomarker for the diagnosis and prognosis of liver cirrhosis and spontaneous bacterial peritonitis [96].

Hence, there is very little published research on the diagnostic value of CLP among different types of PEs. Recent evidence suggests that CLP levels were significantly higher in pleural fluid of malignant PEs than non-malignant ones and complicated parapneumonic PEs than uncomplicated ones. To date, there are no data supporting a prognostic role of CLP in PEs.

## 6. CLP in Cystic Fibrosis and Non-Cystic Fibrosis Bronchiectasis

CF is a heterogeneous multiorgan disease. In that context, fecal CLP is used to assess intestinal inflammation in CF patients; however, recent studies showed a correlation between bowel and lung disease as regards to drug response and CLP kinetics in CF patients [97,98]. Higher fecal CLP levels were found in patients with a severe CF phenotype (Pseudomonas aeruginosa airways colonization, forced expiratory volume in the first second (FEV1) < 50% of predicted, pancreatic insufficiency, underweight status). Consistently, decreased fecal CLP levels have been reported after antibiotic treatment for respiratory exacerbation in CF patients [98,99]. Therefore, fecal CLP can be useful to monitor a clinical worsening longitudinally [100].

In that context, the efficacy of probiotics for improving health outcomes in children and adults with CF has been a much-debated issue [101]. A recent meta-analysis confirmed that probiotics significantly reduce fecal CLP in children and adults with CF; however, probiotics may make little or no difference to pulmonary exacerbation rates. Nevertheless, further evidence is required before firm conclusions can be made [101].

CLP is also found abundantly in CF sputum and serum [97]. When CLP was measured in induced sputum and serum, it could diagnose pulmonary exacerbations [102] and lung function decline in previously stable CF patients [97]. Besides, high sputum CLP levels have been associated with a more rapid recurrence of pulmonary exacerbations [103,104]. Another study reported significantly higher circulating serum CLP concentrations in a cohort of CF patients with severely impaired lung function compared to a CF cohort with mild lung function impairment [105]. However, validation studies are required in order to implement diagnostic thresholds [106]. Besides, sputum and serum CLP levels were found to have variable success in detecting response to inflammatory treatments [99]. For instance, early decreases in serum CLP levels were predictive of response to azithromycin to modify exacerbation risk [107].

On the other hand, the innate immune protein CLP has been found to promote co-colonization of polymicrobial communities and interaction between Pseudomonas aeruginosa and Staphylococcus aureus in murine lung, given that metal fluctuations are a driving force of microbial community structure [108]. Exposure to this host protein known to sequester metal ions at infectious foci recapitulates responses occurring within metal-deplete portions of the biofilm. Moreover, Kang et al. revealed a capacity of S100A8 and S100A9 proteins to induce MUC5AC production in airway epithelial cells, serving as mediators linking neutrophil-dominant airway inflammation to mucin hyperproduction [109].

Furthermore, only one study investigated potential biomarkers of suppurative and inflammatory lung disease in induced sputum samples of patients with non-cystic fibrosis bronchiectasis that reported that calgranulins a, b, and c were highly abundant in patients with non-cystic fibrosis bronchiectasis, along with CF patients, compared to controls [103].

This evidence supports that fecal CLP could monitor systematic CF. At the same time, the increase of serum or sputum CLP levels potentially predicts pulmonary exacerbations and lung function decline in previously stable CF patients, although contradictory data support that CLP promotes co-colonization of polymicrobial communities. The clinical implications of these findings require further investigation. 

## 7. CLP in Asthma

Animal and clinical research has linked CLP with asthma [110]. Existing evidence recognizes the relationship between CLP and type 2–dominant immunity [9]. Likely, CLP influences directly or indirectly multiple cell types to confer protection in the lung upon the Alternaria fungi challenge [111]. Research using neutralizing antibodies for S100A8 and S100A9 showed that CLP promoted disease in a mixed allergen model [110,112]. In CLP-deficient mice, an inability of T regulatory cells to control Th2 responses in increased allergic airway inflammation was shown [9]. Although it was demonstrated that CLP modulates T regulatory cell activation by directly suppressing Th2 cell function, changes in C-C motif chemokine ligand (CCL)11 and CCL24 that promote eosinophilia could also indicate direct or indirect effects of CLP on the airway epithelium [111].

On the other hand, exogenous treatment of S100A8 and S100A9 reduced T helper cell 2 (Th2)-mediated responses after ovalbumin-induced allergic airway inflammation [113,114], suggesting that CLP protects against allergic airway inflammation [113,114]. Previous experimental studies found that S100A8 leads to airway hyperresponsiveness by suppressing airway smooth muscle cell contractility in type 2 allergic airway disease in rats [111], modulating mast cell function, and suppressing eosinophil migration in acute asthma [113].

CLP has been implicated in type 2 allergic airway disease in mice by binding to and activating via TLR4 and RAGE [101,112]. It was found that the localization of TLR4 and RAGE within the lung during allergic exposure could influence CLP-mediated protection [110,112]. TLRs are qualitatively and differentially expressed on the hematopoietic and structural airway or non-hematopoietic stromal cells. When these cells are activated by TLRs agonists, such as a house dust mite allergen, an immune-modulatory role is exerted in asthma development [112].

Although CLP is implicated in the pathogenesis of inflammatory diseases by functioning as a ligand for TLR4 and RAGE, an alternative anti-inflammatory mechanism by which S100A9 influences innate and adaptive immune responses has been recently presented [9]. More specifically, it has been elucidated that S100A9 has a novel role in regulating CD4-positive T-cell responses by restricting the number of IL-13/IL-5–producing CD4-positive T cells in the asthmatic lung.

However, increased serum CLP levels were not found in a study of asthmatic allergic children. This finding is possibly due to the low number of children with ever asthma and equal skin prick test positivity in the groups [115]. Furthermore, its biological function in the immunophenotypes of severe asthma has been subject to considerable discussion [110,111,112,113,114,115,116,117,118,119,120,121,122,123].

S100A9 is localized to neutrophils and bronchial epithelial cells in the airway during neutrophil-dominant allergic airway disease [118]. Similarly, in the lung, S100A8 was expressed by neutrophils and macrophages and was upregulated during acute allergic inflammation [110,112,120,121,122]. Sputum S100A8 and S100A9 levels have been linked to more severe, uncontrolled disease phenotypes [8,103,117,118,119], or neutrophilic endotype [120], acting mainly via RAGE [120]. Specifically, Lee et al. reported that S100A9 sputum levels were higher in severe asthma patients with neutrophil-dominant inflammation compared to eosinophil-dominant or paucigranulocytic groups [118]. The neutrophil sputum count was significantly correlated with S100A9 levels [8,119]. Therefore, S100A9 may initiate and amplify neutrophilic inflammation in patients with uncontrolled, severe asthma and could be used as a biomarker of neutrophilic inflammation in this group of patients [118,119,120].

Selective eosinophilic recruitment during allergic lung inflammation has not been shown to link with CLP expression, although S100A9 is one of the most abundant proteins in the peripheral blood eosinophil proteome [8,121].

Another study found that high degree intestinal inflammation at two months of age, determined as high fecal CLP, predicted asthma, and atopic dermatitis by the age of six years suggesting that early changes in the gut immune system have long-term effects on the development of allergic diseases [116]. High fecal CLP levels were linked to the low abundance of fecal Escherichia coli; thus, a down-regulation of intestinal inflammation in the presence of colonization with Escherichia coli in early life was suggested [116]. The impaired activation of protective IL-10 in monocytes due to the lack of colonization with Escherichia coli could explain the intestinal inflammation associated with high fecal CLP and later risk of asthma and atopic dermatitis [116]. In addition, a molecular biology study in a murine model has been suggested that in utero, vitamin D deficiency can alter lung structure and function by increasing inflammation, contributing to symptoms in chronic diseases, such as asthma, via the expression of the inflammatory molecules S100A9 and S100A8 [123].

Therefore, the role of S100A8/A9 may differ based on the inflammatory context in the asthmatic lung [8]. CLP might protect against allergic airway inflammation involved in type 2–dominant immune response in asthma patients. Its biological function in severe asthma forms is not clear, but it might amplify neutrophilic inflammation in severe asthma forms. At the same time, the research should be focused on diverse immune environments such as type 2 low, type 17–associated, or mixed-type asthma [118,119].

## 8. CLP in Chronic Obstructive Pulmonary Disease

A different expression ratio of S100A8/A9 has been reported between acute and chronic lung diseases such as COPD [124]. Both lower respiratory tract inflammation and low-grade systemic inflammation are implicated in the pathogenesis and progression of COPD [124]. A high S100A8 and S100A9 gene expression and high plasma and BAL [125]. CLP levels have been documented in the lungs of mice exposed to mainstream cigarette smoke [126].

On the one hand, S100A8 protects alveolar epithelial type II cells against cigarette smoke-induced injury and emphysema. S100A8 overexpression rescued cell injury [126], while S100A8 oxidation can downregulate anti-inflammatory processes and cytoprotection [127,128]. Considering this, it has been proposed that targeting S100A8 may present a potential therapeutic strategy against COPD development [127].

In contrast to S100A8, it has been reported that S100A9 signaling contributes to the progression of smoke-induced COPD [125]. A recent study found that the loss of S100A9 signaling reduced cigarette smoke-induced airspace enlargement, lung destruction and remodeling, extracellular signal-regulated kinase (ERK) and c-RAF phosphorylation, and the release of matrix metalloproteinases (MMPs)-3, -9, monocyte chemoattractant protein-1 (MCP-1), interleukin-6 (IL-6), and keratinocyte-derived chemokine (KC) into the murine airways [125].

What is more, S100A9 signaling has been suggested to contribute to the age-related COPD independently of the tobacco-associated neutrophilic inflammation. Recent research has established that the treatment with an immunomodulatory compound preventing S100A9 binding to TLR-4 (paquinimod) to non-smoked, aged animals reduced age-associated loss of lung function [125].

It has also been found that human bronchial epithelial cells isolated from COPD donors secreted more S100A9 than cells from healthy donors or smokers during viral-associated COPD exacerbations [129]. The activity of protein tyrosine phosphatase 1B (PTP1B), a relevant modulator of signaling pathways initiated by the activation of tyrosine kinase receptor superfamily with an anti-inflammatory potential, is desensitized in the lung by prolonged cigarette smoke exposure [129], and that reduced response coincided with enhanced S100A9 secretion.

The multiligand receptor RAGE has been implicated in inflammatory responses in COPD [130]. Li et al. characterized the RAGE-ligands axis as a novel driving force for cigarette smoke-induced airway inflammation in COPD [131]. It has been supported that knockout of RAGE demonstrated a protective effect from mainstream cigarette smoke-induced airway inflammation in mice, possibly via downregulating S100A8/A9 expression and amplifying immune-inflammatory responses [130].

From a clinical point of view, CLP exists in secretions from patients with COPD [55,132]. CLP has been related to neutrophil count and neutrophil-to-lymphocyte ratio in patients with stable moderate to very severe COPD, and higher plasma CLP has been suggested as an independent predictor of increased all-cause mortality [133]. As far as COPD exacerbations are concerned, only one clinical study identified that S100A9 is increased in serum and not in the sputum of COPD patients during exacerbation. The increase in S100A9 and HMGB1 was associated positively with the female sex and negatively with infection status in COPD patients during an exacerbation [134].

Treatment with systemic glucocorticoids has a significant impact on the ability of CLP to predict all-cause mortality [135]. Glucocorticoids inhibit S100A8 and S100A9 expression in murine models [136]. Similarly, glucocorticoids may inhibit CLP expression in humans, although contradictory data are also present [135]. S100A8 was found to be a steroid-refractory mediator of lung neutrophilia through the application of in vivo protein profiling methods [135]. On the other hand, systemic glucocorticoids induce neutrophilocytosis and plasma CLP levels have been positively correlated with neutrophil granulocyte count [135]. It seems that the effect of neutrophilocytocis is more profound than this inhibition [135].

Considering all of this evidence, smoking leads to a higher S100A8 expression as a defense mechanism for protecting from smoke-induced alveolar type II cell injury and emphysema pathogenesis, according to in vitro studies and in vivo animal models. S100A8 might represent a potential therapeutic target for treating neutrophilic lung inflammation in conditions where glucocorticosteroid responses are suboptimal [137]. On the contrary, a direct impact of S100A9-mediated signaling on worsening lung function within the aging lung emerged. From a clinical perspective, CLP levels measured in secretions from COPD patients related to neutrophilic inflammation and suggested to be a predictor of all-cause mortality in corticosteroid-free COPD patients.

## 9. CLP in Lung Cancer

More than 150 years ago, Virchow supported that cancer’s origin was linked to chronic inflammation sites [138]. It is now clear that the pro-tumorigenic inflammation results in DNA damage that could certainly initiate and promote cancer risk [139]. CLP participates in inflammation-induced-cancer and tumor-induced-inflammation [5].

Elevated levels of S100A8/A9 have been found within tumor cells and various human cancers [5]. The protein has different outcomes in malignancy and tumor invasion, whether its source is extracellular or intracellular [5]. Intracellular expression of CLP reduces cancer invasiveness by suppressing adhesion characteristics of cancer cells. At high intracellular concentrations, S100A8/A9 does not affect the apoptosis pathway instead by regulating the epithelial–mesenchymal transition and the mesenchymal–epithelial transitions inducing a reduction in cancer cell invasion capacity [5,140]. On the other hand, high extracellular S100A8/A9 levels have cytotoxic effects by inducing cell death program or apoptosis in various tumor cells [5,141], independently of RAGE or Fas-associated protein with death domain (FADD)-dependent death receptors [5,142,143]. Effective extracellular S100A8/A9 concentrations for promoting apoptosis of tumor cells range from 20 to 250 µg/mL. Conversely, low extracellular CLP concentrations finally promote migration of cancer cells and S100A8 and S100A9 expression knockdown and may increase malignancy and tumor invasion. [140,144].

From a pathophysiological and genetic point of view, CLP has been identified as a potent promoter of tumor-mediated immune remodeling. The S100A8/A9 protein acting via specific receptors on the cancer cell surface and increase the nuclear factor kappa B (NF-kB)-dependent transcriptional activity; thus, actively participating in immune modulation responses and inflammation, which play essential roles in the tumor growth and progression [5,142,143,145,146,147,148,149,150,151,152,153]. More specifically, the activation of NF-kB in tumor cells enhances the expression of several genes (including Ccl7, Cxcl1, Ccl5, Slc39a10, Lcn2, Zc3h12a, and Enpp2) that are implicated in tumor migration, angiogenesis, and formation of pre-metastatic niches [5,148,149].

S100A8 plays a significant role in TLR4/myeloid differentiation factor 2 (MD-2) pathway activation; thus, promoting a tumor growth-enhancing immune microenvironment [150]. Furthermore, the S100A8/A9 receptor neuroplastin-β (A9-NPTNβ) axis in lung cancer has been implicated in the disseminative progression. A newly identified downstream signaling pathway has been described, constituting by tumor necrosis factor (TNF) receptor-associated factor 2 (TRAF2) adaptor, nuclear factor (NF)IA/NFIB heterodimer transcription factor, and SAM pointed-domain containing ETS transcription factor (SPDEF) (TRAF2/RAS-NFIA/NFIB-SPDEF), in linking to the aggressive development of lung cancers [151].

From a clinical perspective, S100A9 expression was statistically higher in lung cancer versus healthy volunteers in exhaled breath condensate (EBC) [154]. It has been supported that early-stage non-small cell lung cancer (NSCLC) patients who had overexpression of S100A9 evaluated by immunohistochemical staining within the cancer cells, exhibited a significantly worse overall five-year survival [155]. Similarly, the degree of mRNA expression of S100A9 was a negative prognostic marker [156], and it has been correlated with tumor differentiation and development in NSCLC patients [157]. S100A8 and S100A9 expression was found significantly higher in cancer tissues than in para-cancer tissues and correlated with tumor differentiation, which may be a potential marker for poor prognosis in NSCLC [158].

It has been proposed that serum CLP in combination with a panel of other biomarkers, including sCD26, matrix metalloproteinases (MMPs)-1, -7, -9, epidermal growth factor (EGF), cytokeratin 19 fragment (CYFRA 21.1), and carcinoembryonic antigen, can identify with high sensitivity stage I NSCLC (94.7%) and 100% small-cell lung cancers [159].

A significant role of the S100A8/A9 axis as regards chemotherapeutic response has been recognized. CD14-positive/S100A9-positive inflammatory monocytes, which are a distinct subset of myeloid-derived suppressor cells suppressing T cells by arginase, inducible nitric oxide synthase (iNOS), and the IL-13/IL-4Rα axis in patients with NSCLC, have been associated with poor response to chemotherapy [158]. Myeloid-derived suppressor cells are also crucial players in establishing the pre-metastatic niche, by releasing exosomal S100A8/A9, contributing to immune suppression, angiogenesis, and metastasis [160]. Bevacizumab (BEV), a monoclonal IgG antibody that inhibits angiogenesis by binding and neutralizing vascular endothelial growth factor A (VEGF-A), leads to the restoration of effective anti-tumor immunity by reducing the circulating S100A9-positive myeloid-derived suppressor cells; thus, extended progression-free survival and protected against brain metastasis in patients with EGFR-mutant lung adenocarcinoma [161].

Hence, CLP is an intelligent molecule whose contribution to cancer cell survival and invasion depends on its concentration or location inside or outside the cells [5]. Many studies support that CLP is a potent promoter of tumor-mediated immune remodeling. A significant role of the S100A8/A9 axis as regards to diagnosis, prognosis, and chemotherapeutic response has been recognized.

## 10. CLP and Metastatic Cancers in the Lung

CLP has been identified as a potent promoter of invasion, and metastasis [5,130]. Systemic cancer spread is preceded by establishing the premetastatic niche, a permissive microenvironment in metastasis’s target tissue [160]. Evidence supporting that S100A8/A9 functions as a sign that attracts cancer cells to specific organs by tumor-derived factors locally and systemically, has proven beneficial in metastatic progression [5,162].

For instance, melanoma cells from distant organs induce S100A8/A9 expression in the lung. Conversely, melanoma cells possessing several receptors that sense the S100A8/A9 ligand are attracted to an enriched S100A8/A9 lung environment [163]. Similarly, S100A8/A9 attracts cancer cells in the lung in prostate cancer murine models [164].

It has been found that although S100A8 was expressed at relatively low levels in the tumor cells, expression was 100-fold higher in the lung and liver, which are common sites of metastasis [165]. Despite the relatively high level of S100A8 expression in the lungs of naïve mice, the level of expression increased further and was significantly elevated after only seven days of tumor growth in a breast cancer model. The same pattern was observed for myeloid-derived suppressor cells [165]. Characterization of myeloid-derived suppressor cells from the lungs revealed that the cells were capable of migrating in a dose-dependent manner toward S100A8 [165]. The expression of S100A8 in the lungs might facilitate the recruitment of myeloid-derived suppressor cells, which may, in turn, aid in establishing a metastatic niche capable of suppressing a localized immune response [165].

There is a positive feedback mechanism between S100A8/A9 and several pro-inflammatory factors adding to tumor progression [5,166]. S100A8/A9 induces cytokine and chemokine expression, and stimulates granulocytes and keratinocytes to further synthesize and secrete S100A8/A9 [5]. S100A8/A9 expression in cancer cells causes enhanced immune cell infiltration, especially by neutrophils [164].

The formation of a tumor-promoting premetastatic microenvironment plays a pivotal role in cancer growth [167,168]. Primary tumor cells release pro-inflammatory mediators such as VEGF-A, transforming growth factor-beta (TGF-β), and TNF-α, IL-1, IL-6, chemokines, MMPs, angiogenic factors, and anti-apoptotic proteins, which lead to selective expression of chemo-attractants S100A8 and S100A9 for tumor cells to home on the pre-metastatic sites [5,139,165,169,170,171,172,173,174]. S100A8/A9 knockdown in the cancer cells reduces MMP-2 and MMP-9 expression that is closely related among other mediators to tumor cell migration and invasion [5]. TGF-β promoted the creation of a premetastatic microenvironment in metastatic breast cancer by modulating specific inflammatory growth factors and cytokines; thus, enhancing the ability of circulating cancer cells to seed the lung [165,167].

In that context, S100A8/A9 may serve as a useful target for anti-metastasis therapy by suppressing the invasive and migratory capabilities of tumor cells [174], such as anti-S100A8/A9 monoclonal antibody [163].

S100A8/A9-sensing receptors including TLR4, RAGE, and other also important receptors that have been recently identified, such as extracellular MMP inducer [EMMPRIN), NPTNβ, melanoma cell adhesion molecule (MCAM), and activated leukocyte cell adhesion molecule (ALCAM) also have the potential to become promising therapeutic targets [163]. S100A8/A9 binding to RAGE promoted lung metastasis playing an important role in breast cancer spread [169]. Both have been proposed as potential anti-invasion targets for therapeutic intervention in breast cancer [169]. EMMPRIN is expressed at the invasive conditions of cancer cells. The interaction between S100A9 and EMMPRIN induced cdc42 activation promoting filopodia formation, migration, and cancer cell polarity [5,175,176,177].

Clinically, the usefulness of an antibody-based single-photon emission computed tomography (SPECT) for detecting S100A8/A9 as an imaging marker for pre-metastatic tissue priming has been emphasized in the literature. It has been shown that the S100A8/A9 imaging signal in the pre-metastatic lung has been correlated with the subsequent metastatic tumor burden in the same organ [160].

Taken together, these data suggest S100A8/A9 is upregulated in the premetastatic lung and there is an emerging role of S100A8/A9 to lung metastatic progression/invasion and cancer growth, signaling via known or newly discovered sensing receptors. The aforementioned signaling pathways constituted potential therapeutic targets.

## 11. CLP in Idiopathic Pulmonary Fibrosis and Other Fibrotic Diseases

S100A9 subunit promotes human embryo lung fibroblast cell growth and induces a RAGE-dependent pro-inflammatory cytokine and collagen secretion via the activation of ERK1/2/mitogen-activated protein kinase (MAPK) and NF-κB pathways [178].

In idiopathic pulmonary fibrosis (IPF) patients, the calgranulin B (S100A9), which is released by activated neutrophils and macrophages over the endothelial surface, may participate in phagocyte adhesion and tissue migration, contributing to the fibrotic process [179,180]. Calgranulin B levels were elevated in BALF of patients with IPF and idiopathic non-specific idiopathic pneumonia (NSIP) with a fibrotic pattern, and showed significant correlations with the functional parameters [179]. These data have been successful in raising awareness of the potential function of CLP in IPF. Indeed, a recent study by Machahua et al. suggested that CLP might be a biomarker for disease severity, and it supported that serum CLP levels were significantly increased in patients with IPF compared with healthy controls and correlate with DLCO and Composite Physiologic Index [181]. Calgranulin B was also significantly higher in advanced IPF and idiopathic NSIP patients with chronic respiratory failure requiring long-term oxygen therapy [10].

Genomic, transcriptomic, and proteomic studies found that S100A8 is a gene with high potential for association with sarcoidosis being involved in regulating immune response, cellular proliferation, apoptosis, inhibition of protease activity, and lipid metabolism [182]. BALF MRP-14 levels were also found elevated in sarcoidosis patients and they were associated with disease severity based on chest radiographic stage [181]. It has been demonstrated that MRP-14 is expressed in granulomas from sarcoidosis patients and stimulates fibroblast proliferation in vitro. Moreover, BALF MRP-14 levels have been inversely correlated with diffusion capacity and forced vital capacity in sarcoidosis patients [180]. Besides, higher CLP levels suggest cardiac involvement in sarcoidosis patients [183].

Besides, high serum levels have been associated with some severe manifestations of rheumatic diseases, such as glomerulonephritis and lung fibrosis [3]. CLP serum levels have been associated with specific and severe systemic sclerosis manifestations, including lung fibrosis [184]. S100A8/A9 levels were significantly elevated in diffuse cutaneous scleroderma patients with lung or kidney involvement compared to controls [184]. In a cohort of Asian patients, serum CLP levels were higher in those with diffuse cutaneous systemic sclerosis than those with limited systemic sclerosis [184]. In Caucasian patients, CLP levels were high only in limited SSc-associated lung fibrosis [185]. Patients with more extended fibrosis were characterized by higher S100A8A/A9 concentrations in BALF [164,186,187]. In systemic sclerosis-related interstitial lung disease, BALF CLP concentration was correlated with imaging findings in high-resolution computed tomography, suggesting its usefulness as a marker of the presence and extent of lung fibrosis [188]. These findings suggest that CLP in systemic sclerosis could identify patients who need accurate screening and close follow-up [3]. Moreover, RAGE is implicated in lung fibrosis development [189], and its expression on the fibroblast cell surface has also been correlated with systemic sclerosis severity [190]. To this end, it has been supported that interstitial lung disease is independently associated with increased fecal CLP levels in patients with systemic sclerosis [188,191], although contradictory results are also present [192].

Hence, higher CLP levels have been positively associated with the presence of idiopathic pulmonary fibrosis or other fibrosing interstitial lung diseases. However, our understanding of this relationship, as well as the diagnostic and prognostic role of elevated CLP levels and the exact source of CLP in these patients, remain unclear.

## 12. CLP in Obstructive Sleep Apnea Syndrome

It is known that increased levels of several inflammatory mediators play roles in obstructive sleep apnea syndrome (OSAS). Torun et al. measured serum CLP levels in a small cohort of OSAS patients and found no significant difference in CLP between the OSAS and normal groups [193]. However, significantly increased CLP values were determined in severe OSAS patients when compared to the other OSAS groups [193]. More recently, Kum et al. confirmed that the serum CLP level could be used as an OSA severity indicator [194].

A potential role for MRP-8/14 as an inflammatory biomarker of endothelial dysfunction, a crucial characteristic of OSAS and a powerful risk-marker for cardiovascular risk, has been suggested [195]. Consequently, serum CLP may serve as a novel and reliable biomarker of cardiovascular risk severity in adult OSAS patients [196,197,198,199,200]. Plasma MRP-8/14 levels were also associated with pediatric OSAS, being suggested to reflect an increased risk for cardiovascular morbidity in children [197,198,199]. In that context, a decrease of post-CPAP treatment serum CLP levels combined with high sensitivity C-reactive protein amelioration could provide evidence for reducing post-CPAP treatment cardiovascular risk [196].

A greater focus on CLP function in OSAS could produce interesting findings that account more for its prognostic role in the disease.

## 13. CLP in Pulmonary Embolism and Pulmonary Hypertension

Up to now, there has been only one study in the literature analyzing the capacity of serum CLP to predict an early postoperative pulmonary embolism in glioma patients compared to those patients who did not undergo surgery [201]. In that study, CPL has been proposed as a risk stratification tool that could help to tailor the thromboprophylaxis in the high-risk subgroup during the perioperative period, allowing a closer follow-up to minimize the incidence and morbidity of venous thromboembolism [201].

Recent evidence suggests a strong association between serum CLP as a marker of hyperactive neutrophils or NETosis and high D-dimer levels in patients hospitalized with COVID-19 [202]. A recent study has implicated CLP in COVID-19 intestinal-related disease pathogenesis [179]. Fecal CLP and D-dimer were positively correlated; this shed new light on the potential role of CLP in thrombosis and the consequent hypoxic intestinal damage [202]. A large case-control study recently demonstrated an increased thrombotic risk in patients with increased CLP levels [203]. At the same time, CLP has been used as a neutrophil activation marker to predict venous thromboembolism in patients with pancreatic adenocarcinoma and extrahepatic cholangiocarcinoma [204].

To this end, relatively little research has been carried out on the role of CLP in pulmonary arterial hypertension. It has only been reported that RAGE and S100A8/A9 are overexpressed in pulmonary artery smooth muscle cells in the absence of any external growth stimulus in patients with either heritable or idiopathic pulmonary arterial hypertension. Nevertheless, further studies are needed to clarify the precise role of S100A8/A9 in the pathophysiology of pulmonary arterial hypertension [205].

Figure 1 illustrates the structure of calcium loaded CLP and summarizes the proposed biological functions.

## 14. Conclusions

The overall literature emphasizes the high importance of prognostic and predictive biomarkers in improving patients’ monitoring, management, and eventually, lung disease control, supporting a treat-to-target approach. This review addresses the emerging involvement of CLP as a danger signal in lung diseases, highlighting the possibility of harnessing and deploying CLP either as a biomarker for the appropriate diagnosis and prognosis of lung diseases, as therapeutic in the treatment of tumors, or as an adjunct to antibiotics for the prophylaxis of infections. In addition to its anti-microbial functions, CLP also serves as a molecule with pro- or anti-tumor properties, which is also involved in pre-metastatic niche formation, depending on its concentration or location inside or outside the cells. The findings of this review potentially introduce CLP in daily clinical practice within the spectrum of respiratory diseases.

## Figures and Tables

**Figure 1 ijms-22-01706-f001:**
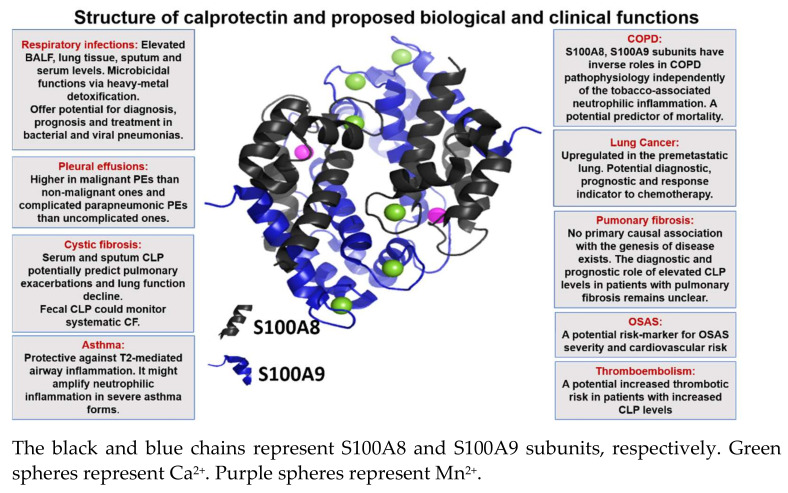
Crystal structure of calprotectin and proposed functions as regards to respiratory diseases.

## Data Availability

The data that support the findings of this study are available on request from the corresponding author, O.S.K.

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
