# Peer review of "Calprotectin in Lung Diseases"

_ijms, 2021, doi:10.3390/ijms22041706_

Round 1

Reviewer 1 Report

Review report of manuscript entitled “Calprotectin in lung diseases” by Kotsiou OS et al

General

Positive:

The authors outline comprehensive literature on Calprotectin (CLP) to update our knowledge on this important neutrophil cytoplasmic constituent. The authors compiled extensive recent reports on CLP and its association with the development of various pulmonary diseases. Many review articles have been written on CLP in bowel diseases, but little on pulmonary diseases. It makes this review interesting to read. In general, this article is well written.

Major Critics:

  1. The authors need to tone down the significance of CLP in various pulmonary diseases. The observations on CLP in various pulmonary diseases discussed in this manuscript are not necessarily causative relation. Therefore, the authors should be cautious in using the word “the roles of CLP in….” (Titles of sections 3-8, 10-13), when the authors discussed CLP in various pulmonary diseases. The observed abnormal serum levels of CLP in pulmonary diseases might not necessarily have primary causal association with the genesis of the diseases.
  2. No discussion on how CLP is released from neutrophils. Considering that it is constituent of neutrophil cytoplasm, it might require disruption of neutrophil cell membrane. If so, neutrophils might require higher degree of stimulation and longer time line to release CLP than other bioactive agents. In earlier stages of neutrophil activation, neutrophils would undergo degranulation whilst keeping the integrity of their cell membrane. In this process, they would release their granular contents, such as gelatinase, myeloperoxidase (MPO), Oncostatin M (OSM), that have also potential roles in inflammation and infection. This is another reason for the authors to be cautious to propose “roles” of CLP in various pulmonary diseases.

Minor Critic:

  1. Lines 87-125 are mostly introduction of CLP. Hence, they should be part of section 1 (Introduction).

Suggestion:

  1. Including cartoons to illustrate the structure of CLP will facilitate the readers at large to better understand the proposed biological functions in relation to its molecular structure.

Author Response

COMMENTS FROM REVIEWER 1

  1. General

Positive: The authors outline comprehensive literature on Calprotectin (CLP) to update our knowledge on this important neutrophil cytoplasmic constituent. The authors compiled extensive recent reports on CLP and its association with the development of various pulmonary diseases. Many review articles have been written on CLP in bowel diseases, but little on pulmonary diseases. It makes this review interesting to read. In general, this article is well written.

RESPONSE: Thank you very much for your kind words about our paper. We are delighted to receive positive feedback from you. We appreciate you taking the time to offer us your comments and insights related to the manuscript. The thoughtful guidance provided by you has helped improve the quality of our manuscript. In the following pages are our point-by-point responses to each of your comments

  1. Major Critics: The authors need to tone down the significance of CLP in various pulmonary diseases. The observations on CLP in various pulmonary diseases discussed in this manuscript are not necessarily causative relation. Therefore, the authors should be cautious in using the word “the roles of CLP in….” (Titles of sections 3-8, 10-13), when the authors discussed CLP in various pulmonary diseases. The observed abnormal serum levels of CLP in pulmonary diseases might not necessarily have primary causal association with the genesis of the diseases.

RESPONSE: We are appreciative of this valuable remark. Per your suggestion, we have revised our paper accordingly and feel that your comments helped clarify and improve the quality of our paper (The titles of sections 3-8, 10-13 have been revised). We also toned down the significance of CLP in various pulmonary diseases throughout the manuscript being more cautious in using the word “roles” (e.g., Abstract: page 1, lines 15-17 and Conclusions, page 23, line 1158-1159).

  1. No discussion on how CLP is released from neutrophils. Considering that it is constituent of neutrophil cytoplasm, it might require disruption of neutrophil cell membrane. If so, neutrophils might require higher degree of stimulation and longer time line to release CLP than other bioactive agents. In earlier stages of neutrophil activation, neutrophils would undergo degranulation whilst keeping the integrity of their cell membrane. In this process, they would release their granular contents, such as gelatinase, myeloperoxidase (MPO), Oncostatin M (OSM), that have also potential roles in inflammation and infection. This is another reason for the authors to be cautious to propose “roles”of CLP in various pulmonary diseases.

RESPONSE: Thank you for this direction. In the revision, we introduced the potential mechanisms of extracellular release of CLP from neutrophils or other cells where it is found, including monocytes and macrophages (page 3, lines 113-145). Please also kindly note that the title of this section (Section 2) has been changed to “The expression of CLP in lung tissue and potential mechanisms of extracellular release of human phagocyte CLP”.

  1. Minor Critic: Lines 87-125 are mostly introduction of CLP. Hence, they should be part of section 1 (Introduction).

RESPONSE: Thank you for this point. The lines 87-102 were transferred to the Introduction section, as suggested (pages 1-2, lines 39-49 in the revised draft). We have also introduced a subsection (Section 3) to summarize the CLP’s concentrations variation in adults and children (pages 4-5, lines 191-218).

  1. Suggestion: Including cartoons to illustrate the structure of CLP will facilitate the readers at large to better understand the proposed biological functions in relation to its molecular structure.

RESPONSE: Thank you very much for your suggestion. A figure (Figure 1) has been added to illustrate the structure of CLP (page 23).

We found your feedback very constructive. We tried to be responsive to your concerns and the manuscript has been massively revised. We hope you find these revisions rise to your expectations.

Reviewer 2 Report

The topic is interesting and important. It is clear that the authors reviewed a lot of the scientific publications that existed at the time of writing the manuscript.

However, the presentation of the information in the manuscript should be significantly corrected. The text should be more clearly structured.

Major concerns:

  • The aim of the authors was to show the place of Calprotectin in disease pathophysiology and in clinical practice. Therefore, it is important to separate or very clearly summarize the pathophysiological and the clinical parts for all discussed diseases – to show the role of Calprotectin and its components for pathophysiological processes and clinical outcomes. At present version of the manuscript these parts are too mixed.
  • The role of Calprotectin and its components should be clearly separated or clearly summarized, especially in the case of clinical aspects (e. g. clinical course of the particular disease, pulmonary function, as prognostic markers for stability or exacerbation).
  • A clearer distinction between the chronic stable phase and the exacerbation phase of a specific discussed lung disease (e.g. tuberculosis, sarcoidosis, COPD, asthma) should be done.
  • A clear distinction should be made between isolated lung diseases and combinations of disease (e.g. cystic fibrosis involving both the lungs and the gut injury).
  • Make a clearer distinction between conditions that may influence neutrophilic inflammation (e.g., smoking vs. non-smoking in COPD, corticosteroid using or not using).
  • Provide only specific information in the sections of the manuscript. Chapters 8, 9, 10, 11, 12 contain too many unrelated commonalities.

Minor concerns:

  • In Chapter 3 authors uses the unusual terminology “hypersensitivity type granuloma”, “non-hypersensitivity granuloma”, “non-immunological granuloma”. In clinical practice histologically granulomas are discriminated into epithelioid giant cells granulomas with or without necrosis. Compact or non-compact granulomas.
  • In Chapter 4 (considering the topic of the manuscript) authors incorrectly uses the terminology “benign PE”. Because "benign PE" is fundamentally different: transudate, exudate (neutrophilic, as in the case of parapneumonic pleurisy; lymphocytic, in the case of tuberculous pleurisy, etc.).
  • In Chapter 10 authors discuss the very different diseases IPF and sarcoidosis together. In the case of sarcoidosis, the term “severity” mentioned in the text is unclear - whether it means active severe sarcoidosis or stage IV (fibrous) sarcoidosis.

Author Response

COMMENTS FROM REVIEWER 2

  1. The topic is interesting and important. It is clear that the authors reviewed a lot of the scientific publications that existed at the time of writing the manuscript.

RESPONSE: We are very grateful for the effort you dedicated to reviewing our submission, as well as your favorable comments.

  1. However, the presentation of the information in the manuscript should be significantly corrected. The text should be more clearly structured.

RESPONSE: We paid heed to your advice and suggestions, and the manuscript has been massively revised. We hope the revised version of our manuscript will satisfy your concerns. The thoughtful guidance provided by you has helped improve the quality of our manuscript. In the following pages are our point-by-point responses to each of your comments.

  1. Major concerns: The aim of the authors was to show the place of Calprotectin in disease pathophysiology and in clinical practice. Therefore, it is important to separate or very clearly summarize the pathophysiological and the clinical parts for all discussed diseases – to show the role of Calprotectin and its components for pathophysiological processes and clinical outcomes. At present version of the manuscript these parts are too mixed.

RESPONSE: We greatly appreciate this valuable comment. We have now significantly revised our manuscript, and we have restructured and rewritten it to state more clearly the pathophysiological and the clinical parts for all discussed diseases according to the available data. At the end of each part, we also provide an objective summary of the section (page 8, lines 360-365; page 11, lines 525-529; page 12, lines 599-603; page 14, lines 680-685; pages 16-17, lines 827-835; page 19, lines 950-954; page 21, lines 1042-1045; page 22, lines 1094-1097; page 22, lines 1116-1117; page 22, lines 1135-1140)

  1. The role of Calprotectin and its components should be clearly separated or clearly summarized, especially in the case of clinical aspects (e. g. clinical course of the particular disease, pulmonary function, as prognostic markers for stability or exacerbation).

RESPONSE: Thank you for this remark. As we have previously stated, in the revision we try to define the clinical aspects of CLP in pulmonary diseases. In that context, we introduced Figure 1 to illustrate the structure of CLP and clearly summarize proposed biological functions.

  1. A clearer distinction between the chronic stable phase and the exacerbation phase of a specific discussed lung disease (e.g. tuberculosis, sarcoidosis, COPD, asthma) should be done.

RESPONSE: You raise a very valid point. We tried our best to improve the manuscript following your direction.

  1. A clear distinction should be made between isolated lung diseases and combinations of disease (e.g. cystic fibrosis involving both the lungs and the gut injury).

RESPONSE: Thank you for this comment. We tried to clearly distinct isolated lung diseases from combinations of diseases (page 7, lines 341-349; page 11, lines 535-543)

  1. Make a clearer distinction between conditions that may influence neutrophilic inflammation (e.g., smoking vs. non-smoking in COPD, corticosteroid using or not using).

RESPONSE: Thank you for this remark. In the revision we try to deal with these issues on pages 15, lines 760-778 and pages 16-17, lines 811-835.

  1. Provide only specific information in the sections of the manuscript. Chapters 8, 9, 10, 11, 12 contain too many unrelated commonalities.

RESPONSE: Thank you for this point. The manuscript has been revised and unrelated commonalities have been deleted. We believe that the contents and the clarity of our draft are much improved in the revised version.

  1. Minor concerns: In Chapter 3 authors uses the unusual terminology “hypersensitivity type granuloma”, “non-hypersensitivity granuloma”, “non-immunological granuloma”. In clinical practice histologically granulomas are discriminated into epithelioid giant cells granulomas with or without necrosis. Compact or non-compact granulomas.

RESPONSE: Thank you for this point. We revised the terminology per your suggestion (page 7, lines 326-337).

  1. In Chapter 4 (considering the topic of the manuscript) authors incorrectly uses the terminology “benign PE”. Because "benign PE" is fundamentally different: transudate, exudate (neutrophilic, as in the case of parapneumonic pleurisy; lymphocytic, in the case of tuberculous pleurisy, etc.).

RESPONSE: Thank you for this remark. We revised the manuscript accordingly on page 10, lines 492-502.

  1. In Chapter 10 authors discuss the very different diseases IPF and sarcoidosis

RESPONSE: Thank you for this comment. In the revision, we discuss separately the very different diseases IPF and sarcoidosis (page 21, lines 1051-1062 for IPF, and lines 1063-1073 for sarcoidosis).

  1. In the case of sarcoidosis, the term “severity” mentioned in the text is unclear - whether it means active severe sarcoidosis or stage IV (fibrous) sarcoidosis.

RESPONSE: Thank you for this point. In the case of sarcoidosis, the term “severity” mentioned in the text refers to disease severity based on chest radiographic stage. We clarify this point on page 21, line 1068.

We are grateful for the time and energy you expended on our behalf. We sincerely appreciate all insightful comments, corrections and suggestions which helped us to improve the quality of our manuscript and carry out a major revision of the paper. We found your feedback very constructive. We hope you find these revisions rise to your expectations.

Round 2

Reviewer 2 Report

Thank you for the corrections.